

# Molecular analysis of oral microflora in patients with primary Sjögren's syndrome by using high-throughput sequencing

Zhifang Zhou[1], Guanghui Ling[1], Ning Ding[2], Zhe Xun[1], Ce Zhu[1], Hong Hua[3] and Xiaochi Chen[4]

[1] Department of Preventive Dentistry, Peking University School and Hospital of Stomatology, National Engineering Laboratory for Digital and Material Technology of Stomatology, Beijing Key Laboratory of Digital Stomatology, Beijing, People's Republic of China

[2] The 3rd Dental Center, Peking University School and Hospital of Stomatology, National Engineering Laboratory for Digital and Material Technology of Stomatology, Beijing Key Laboratory of Digital Stomatology, Beijing, People's Republic of China

[3] Department of Oral Medicine, Peking University School and Hospital of Stomatology, National Engineering Laboratory for Digital and Material Technology of Stomatology, Beijing Key Laboratory of Digital Stomatology, Beijing, People's Republic of China

[4] Department of Oral Biology, Peking University School and Hospital of Stomatology, National Engineering Laboratory for Digital and Material Technology of Stomatology, Beijing Key Laboratory of Digital Stomatology, Beijing, People's Republic of China

Corresponding author
Xiaochi Chen,
chenxiaochi@pkuss.bjmu.edu.cn

## ABSTRACT

**Background**. The objective of this study was to characterize the oral microflora profile of primary Sjögren's syndrome (pSS) patients, thereby revealing the connection between oral bacterial composition and dental caries, and to identify the "core microbiome" in the oral cavities of pSS patients and systemic healthy individuals by using a high-throughput sequencing technique.

**Methods**. Twenty-two pSS patients and 23 healthy controls were enrolled in this study. Their clinical data and oral rinse samples were collected. The V3–V4 hypervariable regions of the bacterial 16S rRNA gene of samples were amplified and analyzed by high-throughput sequencing on the Illumina Miseq PE300 platform.

**Results**. Both two groups were age- and sex-matched. There were significantly higher decayed, missing and filled teeth (DMFT) and decayed, missing and filled surfaces (DMFS) in the pSS group than in the control group ($p < 0.01$). Alpha diversity was depleted in pSS patients, compared with healthy controls ($p < 0.01$), while beta diversity between the two groups was not significantly different. Seven discriminative genera (LDA > 4) were found between the two groups in LEfSe (LDA Effect Size) analysis. The relative abundance of *Veillonella* in pSS patients was fourfold higher, while *Actinomyces*, *Haemophilus*, *Neisseria*, *Rothia*, *Porphyromonas* and *Peptostreptococcus* were significantly lower in pSS patients than in healthy controls. However, the correlation between *Veillonella* and DMFT/DMFS was not significant ($p > 0.05$). In Venn diagram analysis, nine genera shared by all samples of two groups, which comprised 71.88% and 67.64% in pSS patients and controls, respectively.

**Discussion**. These findings indicate a microbial dysbiosis in pSS patients; notably, *Veillonella* might be recognized as a biomarker in pSS patients. The core microbiome in pSS patients was similar to the systemic healthy population. These provide insight

regarding advanced microbial prevention and treatment of severe dental caries in pSS patients. This study also provides basic data regarding microbiology in pSS.

## INTRODUCTION

As one of the most common chronic autoimMune diseases, Sjögren's syndrome (SS) destroys salivary gland tissue, which leads to severe xerostomia (dry mouth) and impairs oral health. The SS prevalence rate is 0.61‰, occurring 10 times more frequently in women than in men (female/male ratio in prevalence data was 10.72); it occurs most commonly in middle-aged women between 40–60 years of age (*Qin et al., 2015*). Primary Sjögren's syndrome (pSS) is marked by a variable degree of activated $CD4^+$ T cell and B cell infiltration (*Christodoulou, Kapsogeorgou & Moutsopoulos, 2010*) of the secretory glands. However, secondary Sjögren's syndrome (sSS) is associated with other connective tissue diseases, such as systemic lupus erythematosus (SLE) and rheumatoid arthritis (RA) (*Vitali et al., 2002*). Xerostomia hinders eating, speaking, and swallowing (all of which largely compromise quality of life for SS patients), and causes rampant caries that always occur in individuals with low saliva flow (*Scully, 1986*; *Gravenmade & Vissink, 1992*; *Lin et al., 2010*).

Researchers have performed several experiments to study oral bacteria in pSS patients. Almståhl et al. cultured and counted the *Lactobacillus* species collected from the supragingival plaque in subjects with hyposalivation (ten subjects with radiation-induced hyposalivation and ten with pSS) and identified them by PCR and restriction fragment length polymorphism (RFLP), and they reported large intra- and inter-individual variations in number and frequencies of Lactobacilli (*Almstahl et al., 2010*). Leung et al. investigated the level of oral micro-organisms in SS patients by using a selective culture technique. They found that the level of *Lactobacillus acidophilus* in stimulated whole saliva ($p = 0.012$) and noncaries associated supra-gingival plaque ($p < 0.0001$) were significantly higher in pSS patients than in healthy controls and sSS individuals (*Leung, Leung & McMillan, 2007*).

Although there have been several studies conducted on the oral micro-organisms of pSS patients, the experimental methods previously used were mostly based on bacterial culture, which is limited to several cariogenic bacteria already known to us. Moreover, traditional methods of culture can only recognize approximately 280 species of oral microbes (*Dewhirst et al., 2010*), while there are more than 700 species in the human oral cavity (*Aas et al., 2005*). Fortunately, a more advanced biotechnique, high-throughput sequencing, has emerged recently, allowing researchers to detect all oral microbes among samples in a single experiment and reveal deeper insights regarding the oral microflora. Siddiqui et al. reported a twofold increase in the relative abundance of *Streptococcus* and *Veillonella* in pSS patients, indicating that microbial dysbiosis was a key characteristic of pSS whole saliva; this dysbiosis was found to occur independent of hyposalivation, after analysis of whole

unstimulated saliva of pSS patients and healthy controls, both with normal salivation, by high-throughput sequencing (*Siddiqui et al., 2016*). With a similar biotechnique, a preliminary study regarding the buccal mucosa microflora of pSS patients found that pSS patients carried a different and less diverse microbial community than that of healthy subjects (*Li et al., 2016*). The pSS group had a greater relative abundance of *Leucobacter*, *Delftia*, *Pseudochrobactrum*, *Ralstonia* and *Mitsuaria* but exhibited reduced abundance of *Haemophilus*, *Neisseria*, *Comamona*, *Granulicatella* and *Limnohabitans*, compared with the control group ($p < 0.05$). De Paiva et al. studied the tongue microbiome of SS patients and found a significant decrease in diversity ($p < 0.05$) and differences in the composition of the microflora in SS patients, compared with their healthy controls (*De Paiva et al., 2016*). An increased relative abundance of *Streptococcus* and a decreased relative abundance of *Leptotrichia* and *Fusobacterium* in the SS group were observed, compared with controls ($p < 0.05$ for all). Though there have been several studies exploring the oral microflora of pSS or SS patients via high-throughput sequencing, the discriminative genera between patient and control groups among the three studies were diverse; notably, none of them discussed the relationship between dental caries and oral microbiota. Thus, the aim of this study was to characterize the oral microflora profile of pSS patients, to identify the "core microbiome" in the oral cavity of pSS patients, and to reveal the connection between oral bacterial composition and dental caries via high-throughput sequencing technique and provide basic data for the oral microbiology of pSS.

## MATERIALS & METHODS

### Subjects

According to the significantly higher SS prevalence rate in women (female/male ratio in prevalence data was 10.72), all patients in our study are female. Twenty-two patients from the Department of Oral Medicine of Peking University School and Hospital of Stomatology were enrolled. The patients were diagnosed with pSS according to the revised international classification criteria (*Vitali et al., 2002*). The control group consisted of 23 age-matched females from a community in Beijing (The Xili community of Xinjiekou Street, Xicheng District, Beijing). All individuals in the control group were systemic healthy and had no xerostomia symptoms. All participants in our study were enrolled who reported that they did not have diabetes mellitus, had not received radiotherapy, had not taken drugs with side effects of dry mouth for 3 months and had no oral mucosal lesion during sampling. They were examined for DMFT and DMFS. Clinical examination and sample collection were performed by a single experienced dentist. The examiner was previously trained and passed the standard consistency test (Kappa > 0.8). All subjects signed written informed consent, and the study passed the ethical review of the Institutional Review Board of Peking University School and Hospital of Stomatology, #IRB00001052-12025.

### Microbial sampling and DNA isolation

Tooth-brushing was not allowed from the night before the appointment until the end of sampling. All subjects were instructed to refrain from eating and drinking for at least 2 h prior to the clinical examination. The examination was performed between 9 A.M. and 11

A.M. The rinse samples were collected into sterile graduated test-tubes after the subjects rinsed their mouths thoroughly for 1 min with 10 ml physiological saline. Samples were immediately transported on ice to the laboratory and centrifuged at 9,000 g for 10 min. The supernatant was discarded and the precipitate was stored at four °C. Within 2 h, the samples were processed following the method of Gao (*Gao et al., 2013*). The precipitate was incubated for 1 h at 58 °C with one ml of lysis buffer (10% sodium dodecyl sulfate (SDS) and 0.2 mg/ml proteinase K in 25 mM Tris–HCl, pH 8), followed by incubation at 80 °C for 10 min to denature the proteinase K. DNA was purified from the lysate by repeated phenol-chloroform-isoamyl alcohol extraction, precipitated with sodium acetate and ethanol, and dissolved in 100 μl sterile Milli-Q water. The final DNA concentration was determined by NanoPhotometer$^{TM}$ Pearl ultramicro ultraviolet spectrophotometer (Implen, Munich, Germany). The DNA quality was checked by 1% agarose gel electrophoresis. A negative control only with buffer was involved during DNA isolation and quantification. The resulting DNA was stored at –80 °C before further analysis.

## High-throughput sequencing

The samples were conveyed to MyGenostics, lnc. (Beijing, China) for high-throughput sequencing. The V3–V4 hypervariable regions of the bacterial 16S rRNA (16S ribosomal RNA) gene were amplified with primers 338F (5′- ACTCCTACGGGAGGCAGCAG-3′) and 806R (5′-GGACTACHVGGGTWTCTAAT-3′) (*Lu et al., 2016*) by PCR (GeneAmp 9700, Applied Biosystems, Foster City, CA, USA), followed by the electrophoresis for quality inspection. Once again, a negative control only with buffer was enrolled during DNA amplification and electrophoresis. The amplified products were sequenced on the Illumina Miseq PE300 platform (Illumina, San Diego, CA, USA).

## Data analysis and statistical methods

The demographic and clinical data were collected and processed via SPSS 20.0. Two-tailed independent-samples $t$-tests, chi-square test and Spearman's rank correlation were used to analyze the data we obtained.

The raw reads obtained by high-throughput sequencing (deposited in the NCBI Sequence Read Archive with accession SRP133569) were processed via Mothur (*Schloss et al., 2009*) and QIIME 1.9.1 (*Caporaso et al., 2010*). The reads were underwent paired-end merging using Mothur with 115 bp in length of overlapping. The de-novo strategy for chimera filtering was employed using VSEARCH (v2.0.2). The quality filtering strategy was performed by truncating those reads shorter than 200 bp in length, those with quality score <20, mononucleotide repeats and homopolymers >10 bp using QIIME. The trimmed and optimized sequences were clustered for operational taxonomic unit (OTU) analysis at a 97% similarity threshold using the QIIME implementation of UCLUST (*Edgar, 2010*). The representative sequences from each OTU were aligned in the Human Oral Microbiome Database (HOMD Release 14.5) (*Chen et al., 2010*) by QIIME.

Bioinformatics analyses, such as alpha diversity (Ace, Chao1, Pielou, Shannon and Simpson indices), beta diversity (PCoA based on weighted Unifrac matrix) were performed in accordance with the OTU table in QIIME. The data of alpha diversity were

**Table 1 Demographic and clinical data of primary Sjögren's syndrome (pSS) patients and healthy controls.**

| Characteristics | Patients ($n = 22$) | Healthy controls ($n = 23$) | $p$ value |
|---|---|---|---|
| Gender (M/F) | 0/22 | 0/23 | |
| Age[a] (Mean ± SD, years) | 61.50 ± 7.47 | 58.00 ± 5.56 | 0.08 |
| No. of individuals with oral mucosal lesion during sampling | 0 | 0 | |
| No. of individuals with any other systemic diseases | 0 | 0 | |
| Radiotherapy | 0 | 0 | |
| No. of individuals taking drugs with side effects of dry mouth for 3 months | 0 | 0 | |
| DMFT[a][**] (Mean ± SD) | 22.95 ± 6.69 | 8.74 ± 5.21 | <0.01 |
| DMFS[a][**] (Mean ± SD) | 74.64 ± 30.52 | 29.04 ± 17.74 | <0.01 |
| Prevalence of caries[b] (%) | 95.45 | 69.57 | 0.06 |

**Notes.**

[a] Independent-samples $t$-tests.
[b] Chi-square test.
[**] $p < 0.01$.

evaluated via independent-samples $t$-tests or nonparametric tests in terms of normality of distribution and visualized by ggplot2 package in R language (v2.2.0). The significance of PCoA was accessed by PERMANOVA test and visualized by ggplot2 package in R language. LEfSe analysis was performed on the online Galaxy workflow framework (http://huttenhower.sph.harvard.edu/galaxy/) (*Segata et al., 2011*); the threshold on the logarithmic linear discriminant analysis (LDA) score was set to 4.0. Heatmap analysis was performed using pheatmap package in R language (v1.0.10). A $p$ less than 0.05 was considered statistically significant.

## RESULTS

### General outline

To characterize the oral microflora profile of pSS patients, and explore the relationship between pSS and caries, we analyzed the sequences of 16S rRNA gene of rinse samples from 45 subjects (22 samples from pSS patients and 23 samples from healthy controls). The average age in both groups was not significantly different ($p = 0.08$). The DMFT and DMFS of patients were significantly higher than those of the control group ($p < 0.01$). The prevalence of caries in patients was 95.45%, whereas it was 69.57% in the control group, but the difference was not significant ($p = 0.06$). These data are presented in Table 1.

After sample detection, all 45 samples were qualified for high-throughput sequencing, including 22 rinse samples of pSS patients (designated as "PR") and 23 rinse samples of healthy subjects (designated as "CR"). We obtained 2579816 raw reads via sequencing. A total of 2,335,289 reads were involved after quality filtration, with an average of 51,895 effective reads per sample. A total of 16,502 operational taxonomic units (OTUs) were observed across all samples and ranged from 357 to 1,852 per sample.

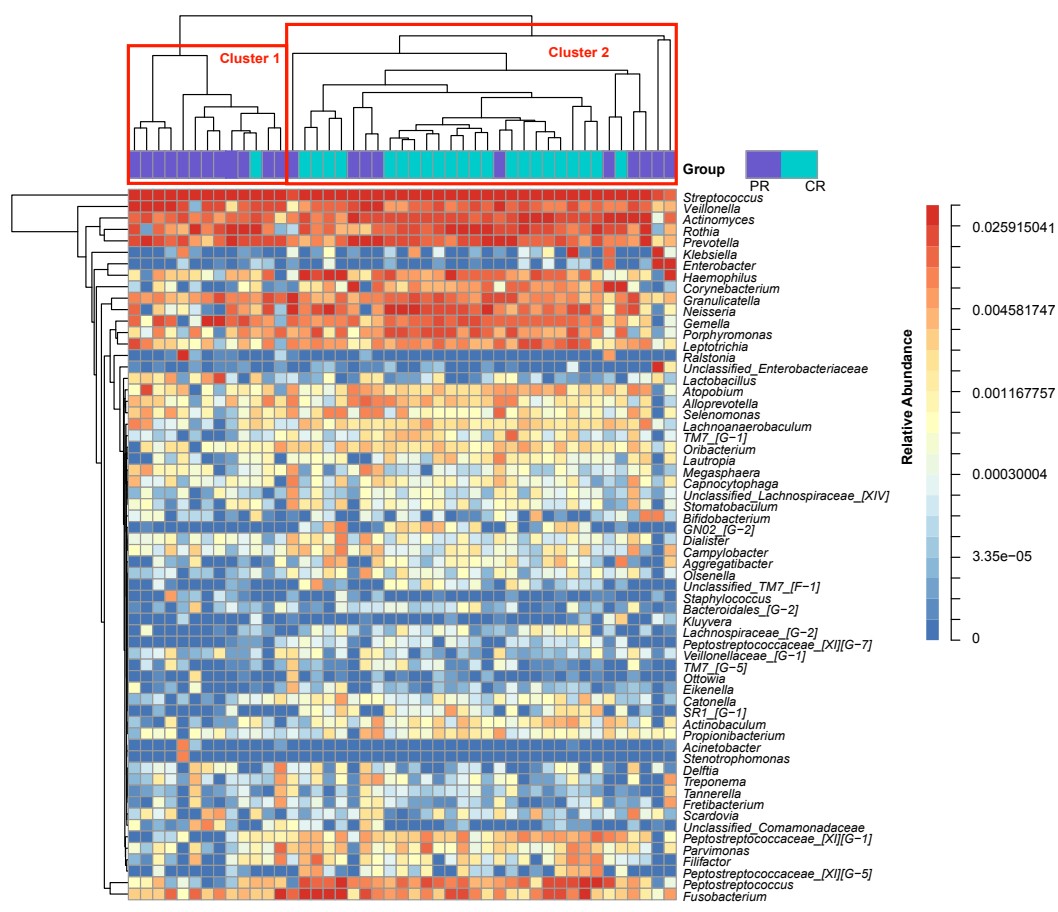

**Figure 1** **Heatmap was clustered based on the microbial community similarity. The genera with relative abundance more than 1% are listed in the column and clustered according to the phylogeny.** PR, rinse samples of pSS patients; CR, rinse samples of healthy controls.

## Global taxonomic features and microbial diversity characteristics in pSS patients and healthy controls

The results of classification and annotation at the phylum and genus levels (only 62 genera with relative abundance >1% are listed) for all samples are shown in Fig. S1. A total of 11 phyla were found across all samples. The dominant phyla included Firmicutes, Actinobacteria, Proteobacteria, Bacteroidetes and Fusobacteria, which comprised 99.33% and 98.82% in pSS patients and controls, respectively. A total of 158 genera were found across samples. *Streptococcus*, *Actinomyces*, *Veillonella*, *Rothia*, *Prevotella*, *Neisseria*, *Haemophilus*, *Granulicatella*, *Fusobacterium*, *Peptostreptococcus*, *Gemella*, *Corynebacterium*, *Klebsiella*, *Enterobacter* and *Ralstonia* were the top 15 most abundant genera, which comprised 87.16% and 84.11% in pSS patients and healthy controls, respectively.

To access the information about the oral microbial communities of pSS patients and healthy controls, we performed the heatmap analysis (Fig. 1). All subjects were divided into two clusters (Cluster 1 and Cluster 2) according to the similarity of samples. All the healthy

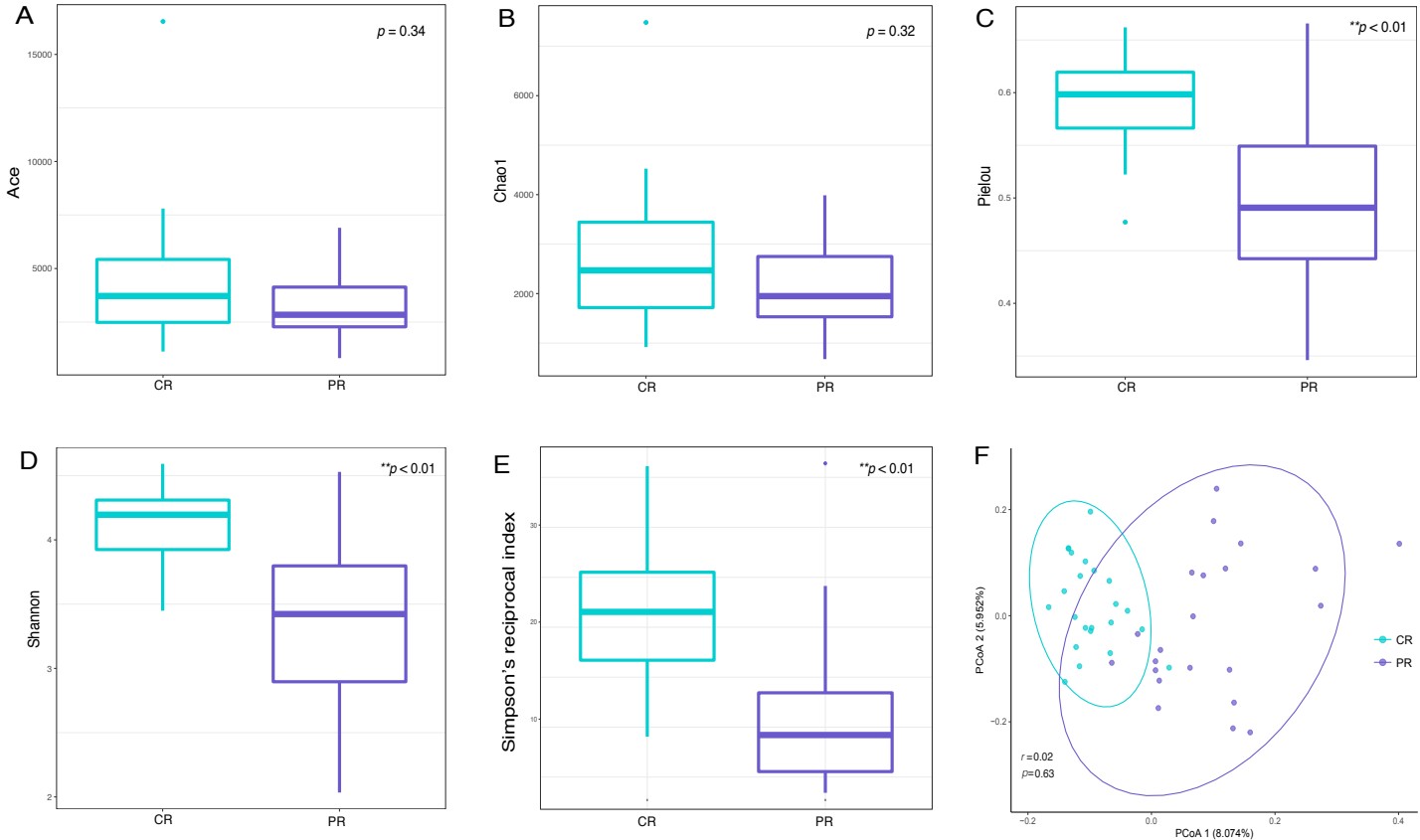

**Figure 2  Comparisons of alpha and beta diversity indices between primary Sjögren's syndrome (pSS) patients and healthy controls.** Higher Ace (A), Chao1 (B), Pielou (C), Shannon (D) and Simpson's reciprocal index (E) indicate higher alpha diversity. Box plots of alpha diversity measure using Pielou (C), Shannon (D) and Simpson's reciprocal index (E) showed a decrease in diversity in the rinse samples from pSS patients ($p < 0.01$). (F) Comparison of principal coordinates analysis (PCoA) of primary Sjögren's syndrome (pSS) patients and healthy controls by using weighted UniFrac distance. Each dot represents one sample and the distance between the samples represents the difference in community composition of the samples. There was no great significance in beta diversity between two groups ($p = 0.63$). ** $p < 0.01$, PR, rinse samples of pSS patients; CR, rinse samples of healthy controls.

controls except one were distributed in Cluster 2. Though pSS patients were relatively scattered, Cluster 1 was almost constructed by them (Fig. 1).

Alpha diversity indices are used to demonstrate the microbial community richness, evenness, and species diversity in sites or habitats at a local scale. Bacterial richness between the two groups was not significantly different, according to Ace and Chao1 indices (Figs. 2A and 2B). However, oral bacterial community evenness and diversity in the pSS group were significantly lower than that in the control group, according to the Pielou, Shannon and Simpson's reciprocal index ($p < 0.01$) (Figs. 2C–2E).

Beta diversity analysis, such as PCoA (principal co-ordinates analysis) via weighted UniFrac distance matrix, was executed to compare the overall bacterial community composition of the two groups. No significant segregation was found between patients and healthy controls (PERMANOVA, $p = 0.63$, Fig. 2F). However, the dots of control group were more concentrated than pSS group as illustrated in the figure.

## Discriminative taxa of pSS patients and healthy controls

The LEfSe (LDA Effect Size) analysis was performed for the exploration of relative taxa abundance, characterized by significant differences between the two groups (i.e., biomarkers). These taxa likely play a crucial role in their local microenvironment; they are listed in Fig. 3. At the genus level, the relative abundance of *Veillonella* in patients (12.22%) was fourfold higher than in the healthy controls (3.21%). However, a relatively higher abundance of *Actinomyces*, *Haemophilus*, *Neisseria*, *Rothia*, *Porphyromonas* and *Peptostreptococcus* was observed in healthy controls, compared with pSS patients. Notably, the well-known cariogenic bacteria such as *Streptococcus* and *Lactobacillus* were found no significant difference between two groups. We performed Spearman's rank correlation to explore the relationship between the relative abundance of *Veillonella* and the severity of caries (Fig. S2), but no great significance was found.

## Core microbiome of pSS patients and healthy controls

The core microbiome is defined as the taxa shared across all samples in a certain group or among groups; it is illustrated by using a Venn diagram (Fig. 4). A total of 158 genera were identified by OTU annotation; 149 genera belonged to pSS patients and 136 to healthy controls. The overlap region A represents the microbiota shared in all samples among patient and control groups, including nine genera: *Leptotrichia*, *Rothia*, *Actinomyces*, *Granulicatella*, *Porphyromonas*, *Prevotella*, *Veillonella*, *Fusobacterium*, *Streptococcus*. There was no other genus shared among pSS patients but an additional 27 genera (region B) were shared among healthy controls, including *Stomatobaculum*, *Dialister*, *Olsenella*, *Capnocytophaga*, *Peptococcus*, *Propionibacterium*, *Bergeyella*, *Megasphaera*, *Selenomonas*, *Lautropia*, *Lachnoanaerobaculum*, *Parvimonas*, *Catonella*, *Alloprevotella*, *Campylobacter*, *Solobacterium*, *Oribacterium*, *Actinobaculum*, *Atopobium*, *Neisseria*, *Corynebacterium*, *Gemella*, *Haemophilus*, *Peptostreptococcus*, *TM7_[G-1]*, *Peptostreptococcaceae_[XI][G-1]*, *Unclassified_Lachnospiraceae_[XIV]*. Thirty-six genera (region A and B) were the core microbiome of healthy controls. Twenty-two genera in region D of PR (*Kocuria*, *Mycobacterium*, *Flavitalea*, *Escherichia*, *Cupriavidus*, *Pedobacter*, *Anaerococcus*, *Cronobacter*, *Ralstonia*, *Stenotrophomonas*, *Burkholderia*, *Rhodocyclus*, *Unclassified_Actinobacteria*, *Unclassified_Alcaligenaceae*, *Unclassified_Lactobacillales*, *Clostridiales_[F-3][G-1]*, *Unclassified_Rhizobiaceae*, *Unclassified_Sphingomonadaceae*, *Unclassified_Carnobacteriaceae*, *Unclassified_Alphaproteobacteria*, *Unclassified_Corynebacteriales*, *Unclassified_Betaproteobacteria*) represented the unique taxa in pSS patients, while nine genera in region D of CR (*Eggerthella*, *Peptoniphilus*, *Sneathia*, *Ruminococcaceae_[G-3]*, *Syntrophomonadaceae_[VIII][G-1]*, *GN02_[G-1]*, *Unclassified_Bifidobacteriaceae*, *Clostridiales_[F-1][G-2]*, *TM7_[G-6]*) represented the unique taxa in healthy controls. The relative abundances of the core microbiome and unique genera are shown in Table 2.

## DISCUSSION

By sequencing and analyzing the 16S rRNA gene of oral microflora of pSS patients and healthy controls, we identified the significant differences in alpha diversity of oral microbial
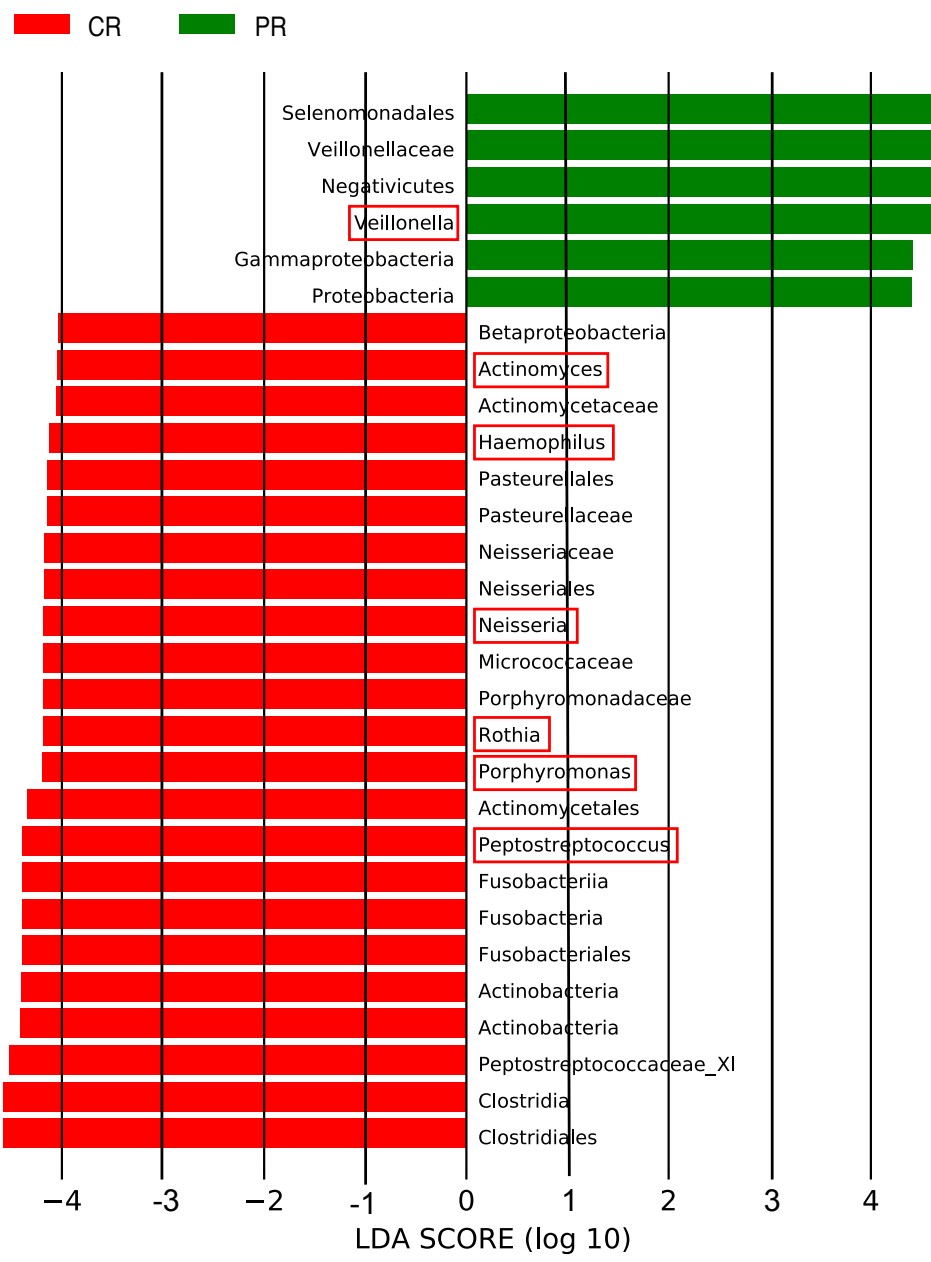

**Figure 3 Comparison of the genera with LDA score > 4 (highlighted by the red box) which calculated by LEfSe analysis between primary Sjögren's syndrome (pSS) patients and healthy controls.** PR, rinse samples of pSS patients; CR, rinse samples of healthy controls.

community and the relative abundance of *Veillonella* between pSS patients and healthy controls, which revealed a dysbiosis in oral microflora of pSS patients and characterized the *Veillonella* may acting as an oral biomarker of pSS.

The rinse sampling method was used in our study. In the context of low salivary secretion in pSS patients and sampling standardization, the rinse technique is more accessible than

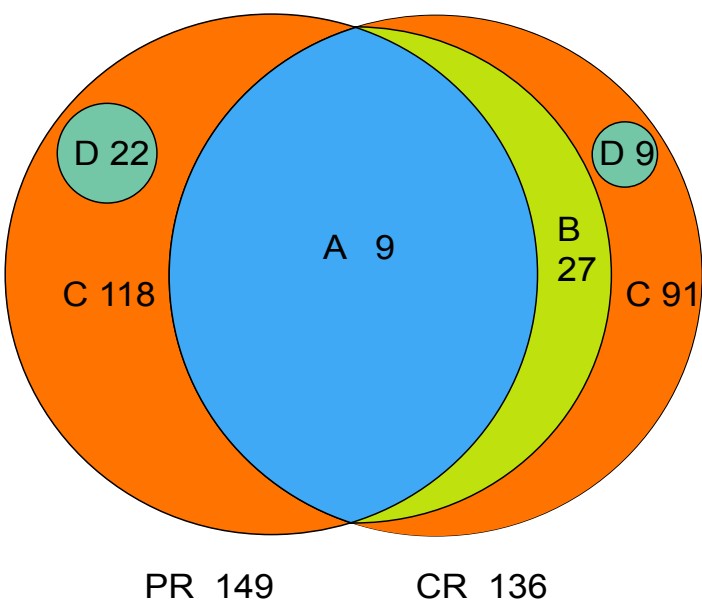

PR 149          CR 136

**Figure 4  Venn diagram.** (A) the genera shared by two groups; (B) the genera shared by CR except the genera in A; (A+B) the genera shared by one group; (D) the unique genera of one group; (C) the genera of one group which belonged to none of A, B and D. PR, rinse samples of pSS patients; CR, rinse samples of healthy controls.

**Table 2  Relative abundance of core microbiome and unique genera in primary Sjögren's syndrome (pSS) patients and healthy controls.**

| Classification | | Genera | Relative abundance (Mean ± SD, %) |
|---|---|---|---|
| A | pSS patients | 9 | 71.88 ± 23.28 |
| | healthy controls | 9 | 67.64 ± 10.26 |
| A+B | pSS patients | 9 | 71.88 ± 23.28 |
| | healthy controls | 36 | 95.90 ± 4.06 |
| C | pSS patients | 118 | 26.83 ± 23.54 |
| | healthy controls | 91 | 4.05 ± 4.02 |
| D | pSS patients | 22 | 1.28 ± 5.69 |
| | healthy controls | 9 | 0.05 ± 0.13 |

**Notes.**
A, the genera shared by two groups; B, the genera shared by CR except the genera in A; A+B, the genera shared by one group; D, the unique genera of one group; C, the genera of one group which belonged to none of A, B and D.

stimulated saliva and plaque sampling and is capable of assessing oral microbiota structure. Several studies have used this method to investigate the oral microbiota. For instance, Almståhl et al. compared the oral microbiota associated with hyposalivation of different origins by analyzing with rinse samples and found the number of *S. mutans* in pSS patients was significantly higher than in the control group with normal salivary secretion ($p = 0.004$) (*Almståhl et al., 2003*). Other studies have suggested that the rinse technique is reliable for measuring oral microflora as well (*Samaranayake et al., 1986*; *Fure & Krasse, 1988*).

More bacteria taxa, even nonculturable ones, can be detected, and the diversity and composition of oral microflora can be studied, by using high-throughput sequencing, in contrast to counting the numbers of selected bacteria by traditional culture methods. Significant differences in oral microbial ecology were present between pSS patients and their healthy counterparts in our study.

The pSS population is known to exhibit higher DMFT and DMFS. A comprehensive study assessing the oral health status of pSS patients reported a higher DMFT among pSS patients compared with healthy controls, regardless of age ($p < 0.05$) (*Christensen et al., 2001*). In addition, a study noted that DMFS in pSS patients was higher than in healthy controls ($p = 0.01$) and DMFS correlated positively with oral dryness ($r_s = 0.53$, $p < 0.05$) (*Pedersen, Bardow & Nauntofte, 2005*). Data from relevant studies also showed that pSS patients had higher DMFT than their healthy counterparts ($p < 0.05$ of both) (*Pozharitskaia & Morozova, 1988*; *Leung, Leung & McMillan, 2007*). Besides the decrease of salivary flow which impaired the functions of oral cleaning and buffering capacity, the change in oral bacteria was another factor to aggravate the development of dental caries. Therefore, the relationship between pSS and cariogenic bacteria was investigated in several studies. An increased level of *S. mutans* and/or *Lactobacillus spp.* was observed in the oral microflora of patients with pSS, compared with healthy subjects, by selective culture (*Almståhl, Wikström & Kroneld, 2001*; Almståhl et al., 2003; *Leung, Leung & McMillan, 2007*). However, no significant differences in *Streptococcus* and *Lactobacillus* were observed between patients and healthy controls in this study. In general, there are possible several reasons for this result. Firstly, though the DMFT and DMFS in healthy controls were significantly lower than in pSS patients ($p < 0.01$), the prevalence of dental caries was high and had no great significance compared with pSS group ($p = 0.06$). Therefore, their oral microflora composition may be closer to the population with high risk of caries which indicated by the increased level of Lactobacillus and cariogenic streptococci, rather than the population with low risk of caries ($p < 0.05$) (*Arino et al., 2015*); this suggests that these bacteria may be incapable of serving as specific biomarkers of pSS patients with a high caries risk. Hence, no obvious differences in the relative abundance of specific cariogenic micro-organisms were observed between the two groups. Secondly, defining the organisms at the genus level, rather than at the species level, is a limitation of 16S rRNA high-throughput sequencing (*Rosselló-Mora & Amann, 2001*). Importantly, in certain genera, the variation of pathogenicity among species can be remarkable. For instance, over 50 species are recognized in *Streptococcus* whereas *S. mutans* is the most prominent cariogenic species among streptococcal population (according to a large body of existing evidence). Other streptococci, known as the "non-mutans streptococci" (*Kawamura et al., 1995*), are predominant in integral tooth surfaces. Even *Streptococcus salivarius*, an oral cavity-based commensal species in this genus, is regarded as a safe and efficacious probiotic capable of maintaining healthy oral microbiota and promoting oral health (*Wescombe et al., 2012*). In consideration of these factors, the relative abundance of cariogenic bacteria in various degrees between the two groups may be perceived if the data is accurate at the species level.

The relative abundance of *Veillonella* in the pSS group (12.22%) was significantly higher than in the control group (3.21%) (LDA score > 4); this result is consistent with several recent studies (*Hayashi et al., 2014*; *Zhang et al., 2015*; *Siddiqui et al., 2016*). *Veillonella* is a Gram-negative anaerobic cocci; its salient physiological characteristic is its lactate-fermenting ability (*Delwiche, Pestka & Tortorello, 1985*), which enables production of propionic and acetic acids, carbon dioxide, and hydrogen (*Rogosa, 1964*). Although the increase of lactate acid led to a raise of *Veillonella* and a decrease in lactic acid, it would eventually hoist the amount of total acid; moreover, there was a symbiotic relationship between *S. mutans* and *Veillonella* (*Noorda et al., 1988*). Another study also showed that the presence of *Veillonella* mainly increased biofilm formation by Streptococcus species (*Mashima & Nakazawa, 2014*). A metagenomic analysis of the bacteria in dental cavities showed that *Veillonella* was one of the most common genera in cavities (*Belda-Ferre et al., 2012*), that was in agreement with a molecular study which also considered *Veillonella* as one of the potential cariogenic bacteria (*Aas et al., 2008*). Notwithstanding, the correlation between *Veillonella* and DMFT/DMFS was not distinct in this study ($p > 0.05$). Therefore, we assumed that the increased level of *Veillonella* might be potentially affected by the disease condition. However, further more rigorous and precise experiments are needed to validate the *Veillonella* acting as a biomarker of pSS in the future.

The relative abundances of *Actinomyces*, *Rothia*, *Peptostreptococcus*, *Haemophilus*, *Neisseria* and *Porphyromonas* were reduced in pSS patients in the present study. *Actinomyces*, together with the non-mutans streptococci, predominates on sound enamel tooth surfaces (*Ximénez-Fyvie, Haffajee & Socransky, 2000*). With the development of dental caries, the microflora switches from dominance by non-mutans streptococci and *Actinomyces* to dominance by *S. mutans* and other non-mutans bacteria, including lactobacilli and *Bifidobacterium* (*Takahashi & Nyvad, 2011*). Similiar to *Actinomyces*, but lesser in abundance, *Rothia* is a frequently observed genus in the initial dental plaque community (*Diaz et al., 2006*) and one of the predominant bacteria in the oral cavity of healthy individuals (*Keijser et al., 2008*; *Bik et al., 2010*). Therefore, it is not surprising that *Actinomyces* and *Rothia* were richer in healthy controls than in pSS patients in our findings.

*Li et al. (2016)* investigated the buccal mucosa microbiota in pSS patients; they reported a lower relative abundance of *Haemophilus* and *Neisseria* in patients ($p < 0.05$). In addition, *Neisseria* exhibited a reduced abundance in subjects with hyposalivation, compared with subjects with normo-salivation, as observed by a team of Japanese scientists (*Hayashi et al., 2014*). In an analysis of salivary microbiota in individuals with different levels of caries experience, a twofold higher relative abundance of *Neisseria* and *Haemophilus* was recorded in the group with low caries experience than in the group with high caries experience (*Belstrom et al., 2017*). Most species in *Neisseria* and *Haemophilus* genera only mildly ferment sugar (*Belstrom et al., 2017*) and exhibit a low acid resistance. As a non-caries-associated species, for instance, *Neisseria subflava* cannot be cultured in the environment where pH is lower than 4.5 (*Bradshaw & Marsh, 1998*). Thus, it is reasonable to speculate that the reduction of non-caries-associated taxa is due to a lower pH environment in pSS

patients, which may result from damage to the buffering function of salivary glands. Though we have not measured salivary or plaque pH in this study. Given these circumstances, it is evident that the pSS patients possess an exceptional bacterial composition and a higher level of dental caries than healthy controls.

There was a significantly lower microbial community diversity in the pSS group than in the control group, consistent with previous reports (*De Paiva et al., 2016*; *Li et al., 2016*; *Siddiqui et al., 2016*). The similarity in richness of the two groups can be attributed to the lower evenness in degree in the pSS group; the results of Pielou analysis confirmed that hypothesis. Despite the disease itself, the huge variation in saliva flow is one of the explanations of the significantly reduced microbial diversity in pSS patients. The saliva maintains a balanced homoeostasis between the highly diverse oral microbiome and oral health status by various mechanisms, such as MUC5B (one of the mucoprotein that is encoded by the MUC5B gene), lysozyme, histatins, β-defensins and the human cathelicidin LL37 (*Van't Hof et al., 2014*). Therefore, decreased saliva flow rate and changed composition that disrupt the biocoenosis of the oral cavity of the pSS patients are crucial factors causing reduced diversity of the oral microflora in pSS patients. Meanwhile, autoimmune diseases and medication can also influence the oral microbiota, and some researchers still suspect a more complex mechanism (*Szymula et al., 2014*; *Zhang et al., 2015*; *Li et al., 2016*; *Siddiqui et al., 2016*; *Van der Meulen et al., 2016*). *Li et al. (2016)* analyzed the impaction of medication on the buccal microflora of pSS patients and suggested that *Streptococcus* and *Lactobacillus* were obviously affected by treatment with prednisone acetas. Another study described a pronounced increase of *Streptococcus* and *Veillonella* in pSS patients with normal salivation, which emphasizes that the disturbance in the biocoenosis of pSS patients can occur independent of hyposalivation (*Siddiqui et al., 2016*). In a study of dental and salivary micro-organisms of individuals with rheumatoid arthritis (RA) launched by Zhang et al., reductions of *Haemophilus spp.*, *Porphyromonas gingivalis* and *Rothia aeria* were observed in the oral cavities of RA patients (*Zhang et al., 2015*). SSA (Sjögren's syndrome Antigen A)/Ro60 is a major autoantigen in SS and SLE (*Schulte-Pelkum, Fritzler & Mahler, 2009*). A recent study with laboratory research revealed that the von Willebrand factor type A (vWFA) domain protein produced by *Capnocytophaga ochracea* was the most potent activator of SSA; further, SSA/Ro60-reactive T cells were activated by recombinant vWFA protein (*Szymula et al., 2014*). These results demonstrate that the autoimMune responses to the normal human microbes and opportunistic pathogens might be the potential trigger of initial autoimmunity in SS and SLE. These observations emphasize that there is change in the oral microbial diversity of patients affected by the disease itself and by medication; inversely, the human microbiota conditions the individual's autoimmunity, which has a strong relationship with hereditary susceptibility. Given the complex and unclear interaction between the oral microbiota and autoimMune diseases, this merits further exploration.

In contrast to the recent study (*De Paiva et al., 2016*), there was no clear difference in the beta diversity of the two groups ($p = 0.63$). Considering that De Paiva et al. reported a significantly lower beta diversity in SS group than in control group ($p = 0.002$) when sequenced the tongue microbiome from ten SS patients and eleven healthy controls, we

speculated that the design of subjects rather than the sample size attributed to this result: the earlier study reckoned without the impact of dental caries, whereas we took that factor into account.

A total of 149 genera in pSS patients and 136 genera in healthy controls were found and nine genera overlapped in Venn diagram. Shared genera are common in the human oral cavity, comprising 71.88% and 67.64% of the relative abundance of microbiota of patients and controls, respectively. Aas et al. amplified and sequenced the bacterial 16S rDNA from nine oral sites (including tongue dorsum, buccal epithelium, supragingival and subgingival plaque and so forth) of five healthy adults (*Aas et al., 2005*). They found that the genera *Gemella*, *Granulicatella*, *Streptococcus*, and *Veillonella* were present in all sites. Xiao et al. sequenced the V1–V3 hypervariable region of bacterial 16S rDNA of supra-gingival plaque of 160 subjects with various levels of dental caries; 99 shared genera in all samples were considered as the core plaque microbiome in those adults. Six genera with the highest relative abundance were *Capnocytophaga* (17.8%), *Prevotella* (13.5%), *Actinomyces* (13.0%), *Corynebacterium* (8.9%), *Streptococcus* (6.6%) and *Neisseria* (6.4%) (*Xiao et al., 2016*). To define the healthy core microbiome of the oral microbial community, Zaura et al. sequenced the microflora of five intraoral niches in three healthy adults; 387 OTUs were present in all samples (*Zaura et al., 2009*). Among these, the abundant taxa (relative abundance more than 0.5%) belonged to *Streptococcus*, *Corynebacterium*, *Neisseria*, *Rothia*, *Veillonellaceae*, *Actinomyces*, *Granulicatella*, *Porphyromonas* and *Fusobacterium*. Therefore, nine genera, which were shared in two groups in the present study, are considered as part of the core microbiome of human oral cavity.

The core microbiome of healthy controls consisted of 36 genera, sharing 95.90% of the abundance. Compared with nine shared genera of patients, the oral microflora of the controls was less variable; the structure of the bacterial community of samples in the control group was more similar to each other for almost all of them were assembled in the Cluster 2 (Fig. 1); that was in general agreement with our results regarding beta diversity, as well as other studies (*Zaura et al., 2009*; *Yang et al., 2012*; *Xiao et al., 2016*). These findings add weight to the hypothesis of the core microbiome in a healthy population. The number of core taxa in patients (nine genera, 71.88%) was less than in healthy controls, implying microbial dysbiosis in the oral cavity of patients.

Twenty-two unique genera were detected in the rinse samples of pSS patients; most of the species, such as *Kocuria rosea*, *Cupriavidus gilardii* and *Anaerococcus prevotii*, are opportunistic pathogens that can cause infectious processes (*Savini et al., 2010*; *Chiu & Wang, 2013*; *Tennert et al., 2014*; *Dotis et al., 2015*; *Du et al., 2015*; *Zhang et al., 2017*), particularly in immunocompromised hosts. Though the pathogenicity of such species is known to us to some extent, the relevance to pSS and dental caries remains undetermined. Nine unique genera appeared in controls, which comprising only 0.05% of known microbes. The poor acid resistance and/or environmental sensitivity of these taxa might be the causative factors of deterioration in oral cavities with low saliva flow rates and the severe level of caries in pSS patients.

## CONCLUSIONS

In conclusion, we observed a clear oral microbial dysbiosis among severe dental caries of pSS patients and *Veillonella* may serve as a biomarker in pSS individuals. The core microbiome of pSS patients was analogous to systemic healthy populations. The results suggest that targeted clinical methods against oral microbes should be considered to prevent the development of severe dental caries in pSS patients.

## ACKNOWLEDGEMENTS

We appreciate the technical support from MyGenostics, lnc. (Beijing, China).

### Funding
The authors received no funding for this work.

### Competing Interests
The authors declare there are no competing interests.

### Author Contributions
- Zhifang Zhou performed the experiments, analyzed the data, prepared figures and/or tables, authored or reviewed drafts of the paper, approved the final draft.
- Guanghui Ling and Ning Ding performed the experiments, contributed reagents/materials/analysis tools.
- Zhe Xun and Ce Zhu authored or reviewed drafts of the paper, approved the final draft.
- Hong Hua conceived and designed the experiments.
- Xiaochi Chen conceived and designed the experiments, approved the final draft.

### Human Ethics
The following information was supplied relating to ethical approvals (i.e., approving body and any reference numbers):

The study passed the ethical review of the Institutional Review Board of Peking University School and Hospital of Stomatology, #IRB00001052-12025.

### Data Availability
NCBI Sequence Read Archive SRP133569.

### Supplemental Information
Supplemental information for this article can be found online at http://dx.doi.org/10.7717/peerj.5649#supplemental-information.

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
