# Peer review of "Molecular analysis of oral microflora in patients with primary Sjögren’s syndrome by using high-throughput sequencing"

_PeerJ, doi:10.7717/peerj.5649_

## Round 0.1 · original submission · Major Revisions

Your article has been reviewed by three experts, who have provided detailed feedback on your submission. There is a significant amount of methodological information missing from the paper. As highlighted by one of the reviewers, full details of methods and bioinformatics tools used in the study should be given. In addition, p values should be corrected for multiple correction testing where appropriate (e.g. using Bonferroni or Benjamini-Hochberg correction). Given you have worked with low-biomass samples, it is imperative that negative control information is provided for this study.

The rarefaction curves shown in Figure 2 add nothing to the manuscript. The information they contain can be adequately described in one or two sentences in the text of the manuscript. One of the reviewers has provided detailed guidance regarding analyses/figures that would improve the article. You should follow this guidance.

Your manuscript will be sent for review again should you choose to submit a revised manuscript.

·

Basic reporting

The authors have provided a clear unambiguous and professional manuscript, that has been well researched and appropriately referenced.

The article is appropriately structured, including correctly labeled figures and tables. The authors have made all data available, however the accession number submitted in the manuscript, reads SRP133569 instead of SRP133529.

The authors have clearly reported a number of appropriate results in the results section however some results not mentioned in the results section are reported in the discussion. It would be appropriate for the authors to report all results within this section, specifically their investigation of differences in specific bacteria, namely Streptococcus and Veillonella in the results.

The authors aimed to reveal any connections between the oral composition and dental caries. Data shown in previous studies was backed up by comparing pSS patients to healthy controls, however, data includes the severity of dental carries in the form of DMFTs and DMFSs, it would be important to explore how the bacterial community changed with the severity of dental changes not just the changes between the 2 groups of patients. This would also allow the validity of Veillonella to serve as a biomarker for disease.

Experimental design

This article is within the aims and scope of the journal and the authors clearly define the knowledge gap they are investigation.

The authors were careful in patient selection however no mention was made of controls being included in sequencing. The inclusion of negative controls or mock communities would be important to confirm there was no contamination. If this was performed, please include the data.

The region of the 16S rRNA gene sequenced was longer than usual for paired end sequencing of 16S, information on how the forward and reverse reads were paired would be useful.

Which chimera checking algorithm was used through mothur?

Validity of the findings

The authors have thoroughly researched the literature and appropriately sited appropriately throughout the paper. No previous publication appears to have investigated the microbiota in SS and its association with dental carries. This appears to be an interesting area as the authors show significantly more decayed, missing or filled teeth or surfaces were observed in those with pSS compared to healthy controls.

The authors carefully explored differences between the pSS patients and healthy controls, however failed to further investigate any correlation between the bacterial community and the dental carries. In order to further added to the field and answer the hypothesis, it would be important to explore how the bacterial community changed with the severity of dental changes not just the changes between the 2 groups of patients. This raw data is presented in the attached demographic and clinical data however this investigation does not appear to have been performed.

It is hypothesised that Veillonella may serve as a biomarker in pSS, is Veillonella increased in pSS patients with more severe carries? A box plot or other appropriate diagram to show the range of Veillonella abundance within the groups would be useful.

Additional comments

1. Line 313-315: The authors assume that an increase in Veillonella increases dental carries, does the data back this up within the pSS group?

2. Please clarify line 284 – 289, It is unclear which study you are referring to.

3. Previous studies have measure oral dryness, was this examined in your patients

The discussion of pH was extremely interesting and would be very interesting to investigate further. This may not be possible for the current study but is a very useful and important point.

Reviewer 2 ·

Basic reporting

Methods are not described to the detail that allows others to repeat the study. some parts, eg, lines 117-119, are incomprehensible due to lack of explanation (in this case, which females, SS patients?) and placement in the text (right after the description of the control group).
Figure legends not complete (eg, fig 1 misses description on what is CR and RR group). Supplementary table on demographic and clinical data - no explanation on what is reported in all those columns starting with C. Figure 4 legend is unclear and should be improved.

Experimental design

Regarding methods, more detail is needed in the part about subjects.
Caries prevalence should be defined. Does this mean only decayed, or the entire DMFS index, thus including filled and missing surfaces?
Was power analysis performed? How is sample size justified?

Validity of the findings

lines 172-173: referring to the question about caries prevalence, in supplementary table on demographic and clinical data, there is one individual in the SS group (assuming that 0 stands for SS group, which again, is not clear) who has 0 decayed surfaces. Nevertheless, the results part in lines 172-173 states that SS group had 100% caries prevalence. Do authors account fillings and missing teeth as caries? That is called 'past caries experience' (assuming that the extractions were performed due to caries, and the same goes for the fillings).
Throughout the results part (and also in the abstract), authors should avoid using terms as 'much lower' , but, instead, report, if the difference was statistically significant, and report the p value.
The same goes for descriptions such as 'relatively concentrated tendency' (line 216 - 'the structure was more conservative' (line 409-410). Please use more scientific descriptions.
Avoid using terms suggesting that this was a longitudinal study, eg, 'clear change in microbial ecology was present' (line 269), 'there was a sharp decline in diversity..' (line 345) etc.
Figure 3: thisfigure does not add any useful information for the reader. Instead, variation in the data (standard deviation per taxon) should be presented.
lines 381-386: here the low sample size should be discussed as a potential reason for lack of difference. Judging from the supplementary figure S3, the beta diversity is quite different, just the groups are too small to get this statistical.
Discussion on Veillonella should be carefully revised - the metabolic products produced by Veillonellae are less cariogenic than lactate (higher pK acid than acetate or propionate) - the energy source and substrate for their metabolism. Increase in lactate production by streptococci is accompanied by increase in Veillonella population and lactate reduction. Most likely link with higher caries prevalence is that absolute numbers of these organisms are higher, thus also the streptococci, resulting in more total acids. Salivary role in clearance of acids has not been mentioned, but is a known reason for caries.

Reviewer 3 ·

Basic reporting

Article “Molecular analysis of oral microflora in patients with primary Sjögren's syndrome by using high-throughput sequencing” study the microbiome in Sjögren's syndrome compared to healthy subjects. The topic of the work is interesting and well-balanced disease and control subjects using 16S molecular techniques. However, some improvements should be addressed

Experimental design

Introduction
The author mentioned line 67 “Researchers have performed several experiments to study oral bacteria in pSS patients” however they didn’t cite the most relevant articles in the last years regarding oral microbiology and Sjögren's syndrome.

Material and Methods
Authors indicated received radiotherapy or treated with drugs. Due to the low number of disease subjects, should be clarified the clinical data. I suggest including a table by subject following info: drugs, chemotherapy treatment, dmfts, oral lesions present during the sample collection, salivary flow rate and demographics such as gender and age.

Bioinformatics analysis was not reported the software used. If the authors used R, should include the packages used (e.g., vegan, ape…) in the methods and the references sections.

I didn’t find negative controls to discard contamination during experimental procedures. Did the authors collect samples for negative control evaluation? Please, indicate and submit the sequences of the negative control samples (extraction and PCR controls)

Validity of the findings

Results
Section 3.1 and 3.2 should be compressed in one, excluding the paragraph line 188-198 (first paragraph of the microbiome analysis section).

Table 1 should include more clinical info relevant to the results such us, flow rate, drugs or radiotherapy treatment.

The authors could start the results describing the cohort selected and explaining the specific clinical characteristics related to the goal of the study, justifying the relevance of caries in the disease cohort.

Paragraph lines 183 to 187 and Figure S1 are not necessary to evaluate the results. I suggest removing the paragraph and Figure S1

Sections 3.3 and 3.4 starting from paragraph line 188-198 should be compressed in one describing the bacterial composition of the cohort.
I suggest revaluating figures to be more informative and precise for the reader to understand.
- The authors could include as Figure 1 Heatmap including individual samples might show the cluster of the most abundant genera in healthy or disease (as they mentioned Veillonella after).
- Figure 2 could be boxplot grouped by Healthy vs. Sjogren for Fig. 2A Richness and Fig. 2B Diversity that would be a more clear visualization of the data to evaluate the differences among study groups
- Figure S3 should be part of Figure 2 as Figure 2C using PCoA plot. Please, evaluate using PERMANOVA the groups CR and PR centroids to reanalyze the statistical significance of the groups, and indicate the percentage of the axis PC1 and PC2

Next section is the most relevant of the study should be developed in more detail, due to the authors found Veillonella as the most pertinent genus related to the disease. Did consider the authors to have a more in-depth analysis of the genus, by species level (e.g., OTUs of Veillonella found or qPCR of the main spp.)?

Section core microbiome could be shown by the heat map (suggested as figure 1) and do a 16S hierarchical clustering of the samples. Some of the genera are related to contamination in other studies, please evaluated by negative controls those findings are real taxa of the samples.

Discussion
The first paragraph should include a resume of the cohort, findings, and conclusion of the study. Please add the section.

Caries is related to some bacterial species, but the authors highlight S.mutans. However, one of the significant findings of this study is Veillonella. Sample chose to rinse not a plaque, so others oral niches could be represented as gingival, mucosal, tongue among others, implying multiple oral manifestations beyond caries that the authors didn’t evaluate.

Lines 333 to 335 is not clear what the authors were claiming. Please, rephrase.

The authors mentioned pH (line 338), decreased saliva (line 354) and medication (line 357) as a possible explanation for the disease microbiome. Did the author measured the pH, salivary flow rate and collect medication info on the subjects? If they did, please include as part of table 1.

Additional comments

They have a good potential in their study, but the authors should consider reanalyze the micriobiome data to cover better their conclusions.

---

## Round 0.2 · accepted · Accept

Your article has been reviewed by two referees, both of whom are satisfied with the extensive revisions you have made to the work. I am, therefore, happy to inform you that your article has been accepted for publication. One of the reviewers has asked for some minor corrections to be made to the manuscript before its publication. Please see text that I have pasted below, and submit an updated manuscript file to the editorial office so that publication of your article can proceed.

There are some very minor errors that should be corrected prior to publication, while in prodiction:

157: "reads were underwent paired-end *reads merging using Mothur" *extra reads not required
161: OTU - operational taxonomic unit
216: should read "matrix"
Fig1: "Heatmap *that all samples" *of

Fig2: Diversity measures do not correlate with higher or lower alpha diversity they indicate or describe the community.
Higher Simpson index (D) does not correlate with lower alpha diversity, it indicates or describes lower alpha diversity. Some researchers find it helpful to use 1-D (Simpsons index of diveristy) or 1/D (Simpsons reciprocal index) to prevent confusion.

"Box plots of alpha diversity measure using Ace (A), Chao1 (B), Pielou (C) and Shannon (D) and Simpson's index (E) show a decrease in diversity in the rinse samples from pSS patients."

Legend should indicate the PCoA was carried out using weighted UniFrac distance.

# ·

Basic reporting

The authors have successfully addressed the issues raised in the first review.

There are some very minor errors that should be corrected prior to publication:
157: "reads were underwent paired-end *reads merging using Mothur" *extra reads not required
161: OTU - operational taxonomic unit
216: should read "matrix"
Fig1: "Heatmap *that all samples" *of

Fig2: Diversity measures do not correlate with higher or lower alpha diversity they indicate or describe the community.
Higher Simpson index (D) does not correlate with lower alpha diversity, it indicates or describes lower alpha diversity. Some researchers find it helpful to use 1-D (Simpsons index of diveristy) or 1/D (Simpsons reciprocal index) to prevent confusion.

"Box plots of alpha diversity measure using Ace (A), Chao1 (B), Pielou (C) and Shannon (D) and Simpson's index (E) show a decrease in diversity in the rinse samples from pSS patients."
Legend should indicate the PCoA was carreid out using weighted UniFrac distance.

Experimental design

Could you please indicate what length of overlap you used to pair the reads.

Validity of the findings

Thank you for adding in the extra analysis, this makes the manuscript much clearer and more informative.

Additional comments

Thank you for your clear response and addressing the issues raised by the reviewers.

Reviewer 3 ·

Basic reporting

The changes made the article clearer. The authors changed all based on my suggestions. I Think the paper is very relevant.

Experimental design

Negative controls, clinical data, and bioinformatic analysis details provide more reproducibility and clinical significance of the work

Validity of the findings

Robust Microbiome findings and statistically supported

Additional comments

I appreciate the authors made the changes that I suggested